# Self-Supervised Rule Learning to Link Text Segments to Relational Elements of Structured Knowledge

**Shajith Ikbal**[†][*], **Udit Sharma**[†][*], **Hima Karanam**[†][*], **Sumit Neelam**[†], **Ronny Luss**[‡],
**Dheeraj Sreedhar**[†], **Pavan Kapanipathi**[‡], **Naweed Khan**[§], **Kyle Erwin**[§],
**Ndivhuwo Makondo**[§], **Ibrahim Abdelaziz**[‡], **Achille Fokoue**[‡], **Alexander Gray**[‡],
**Maxwell Crouse**[‡], **Subhajit Chaudhury**[‡], **Chitra K Subramanian**[‡]

**IBM Research**

## Abstract

We present a neuro-symbolic approach to self-learn rules that serve as interpretable knowledge to perform relation linking in knowledge base question answering systems. These rules define natural language text predicates as a weighted mixture of knowledge base paths. The weights learned during training effectively serve the mapping needed to perform relation linking. We use popular masked training strategy to self-learn the rules. A key distinguishing aspect of our work is that the masked training operate over logical forms of the sentence instead of their natural language text form. This offers opportunity to extract extended context information from the structured knowledge source and use that to build robust and human readable rules. We evaluate accuracy and usefulness of such learned rules by utilizing them for prediction of missing kinship relation in CLUTRR dataset and relation linking in a KBQA system using SWQ-WD dataset. Results demonstrate the effectiveness of our approach - its generalizability, interpretability and ability to achieve an average performance gain of 17% on CLUTRR dataset.

## 1 Introduction

Relation linking is a key step in many Natural Language Processing (NLP) tasks including Semantic Parsing, Triple Extraction, and Knowledge Base Question Answering (KBQA). Its goal is to accurately map predicate components of a natural language (NL) text segment onto their corresponding predicate (i.e., relational) elements within a knowledge base (KB). Such a mapping would enable question answering systems to exploit the deep

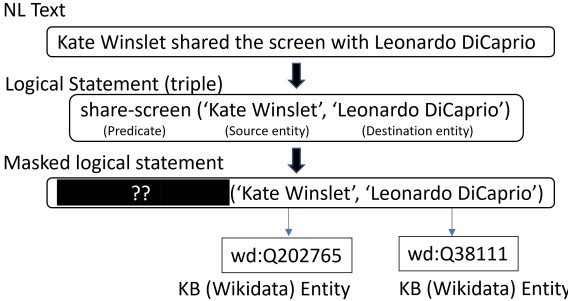

Figure 1: Illustration of logical statement of a NL text, masked logical statement, and the grounding (i.e. linking) of unmasked elements of the statement to a knowledge base (Wikidata).

reasoning capabilities of symbolic systems. For example, by accurately mapping predicate component of the NL text shown in Figure 1, i.e., *share-screen*, to its corresponding relational elements within KB, we should be able to successfully utilize KBQA systems to answer questions like *Whether Kate Winslet shared screen with Leonardo DiCaprio?* and *Who all have shared screen with Kate Winslet?*[1]. However, learning these mappings is challenging for various reasons: lexical variations, modeling deficiencies to handle complex linking involving multiple KB predicates[2], and limited training data to build models in a supervised fashion.

Various approaches have been tried in the past work, including graph-based approaches (Pan et al.,

---

[†]IBM Research, Bangalore/Gurugram, India.
[‡]IBM Research, T. J. Watson Research Center, Yorktown Heights, New York, United States.
[§]IBM Research, Johannesburg, South Africa.
[*]Correspondence e-mails: shajmoha@in.ibm.com, udit.sharma@in.ibm.com, hkaranam@in.ibm.com

[1]Provided entity mentions in the NL text are also appropriately mapped to their corresponding entity elements in the KB, through entity linking (Wu et al., 2019). For example, in Figure 1, entity mention *Kate Winslet* in NL text is mapped to corresponding KB (Wikidata) entity *wd:Q202765*.

[2]On several occasions, there is no single relational element but multiple connected relational elements within the KB that actually correspond to a NL text predicate. These connected relational elements constitute a path within the KB. For example, two connected relational elements as in Figure 3 (i.e., one-hop path involving a reverse KB relation edge *cast-member* and a forward KB relation edge *cast-member*) correspond to NL text predicate *share-screen*. Note that a formal expression of this KB path is given in Figure 2, i.e., $cast\text{-}member^{-1}(x, z)$ & $cast\text{-}member(z, y)$.

Rule: share-screen ← w1*(P161- & P161+)

↓

cast member⁻¹ (x, z) & cast member (z, y)

Description of the rule: x has shared screen with y if
x is a cast member of a movie z where y is also a cast member

Figure 2: Illustration of human readable rule for the example shown in Figure 1 and the corresponding extended knowledge base context shown in Figure 3. Note that inverse knowledge base predicate (i.e., reverse knowledge base relation) in the rule is denoted by superscript $^{-1}$, as described in Section 3.2.

2019; Dubey et al., 2018), linguistic features based approaches (Sakor et al., 2019a,b; Lin et al., 2020), and recently neural approaches that achieve state-of-the-art performance (Rossiello et al., 2021; Mihindukulasooriya et al., 2020; Yu et al., 2017). However, these approaches have the following drawbacks: they are (a) data hungry requiring huge amount of supervised data for training, (b) non-interpretable by design, and (c) non-trivial to adapt and generalize to new domains or knowledge graphs. In this paper, we propose a neuro-symbolic approach that formulates relation linking as a *rule learning* problem in the space of structured knowledge sources. Our approach tries to learn human readable rules of the form illustrated in Figure 2 and use them to perform relation linking, thus resulting in explainable relation linking model. In contrast to the existing approaches that build linking models in a supervised fashion, we adopt a *self-supervised learning* approach that offer benefits: does not require annotating large amounts of training data, and offers generalization across domains.

For self-supervision we use *masked training*, a popularly used self-supervised training strategy for large language models. However, in contrast to the use of NL text as in the large language models (LLMs), masked training in our approach operates on logical statements of the sentences. In this work, we use triples[3] extracted from semantically parsed output of the sentences as the logical statements. An example sentence in its NL form and its corresponding triple form (logical statement) are shown in Figure 1. The masking procedure when applied on a triple may end up masking any of the three elements, i.e., subject, object, or predicate. Figure 1 shows an illustration of the predicate component

getting masked. Given that the three elements of a triple play different semantic roles, the masked training need to be adapted to suit the role played by the element being masked. As will be seen later in the paper, such differences reflect mainly on the loss computation during training.

The main contributions of our work are:

1. The first self-supervised rule learning approach that uses an adaptation of masked language objective for logical forms extracted from text. We have applied this to relation linking, a prominent task for understanding natural language.

2. The output of our model is completely interpretable; the set of rules obtained to perform relation linking can explain the predictions of the model. An example of a learnt rule for the SWQ dataset is shown in Figure 2.

3. We evaluate our approach on two datasets CLUTRR and SWQ demonstrating (a) Generalizability over two different knowledge bases as context on the datasets, (b) Effectiveness of relation linking where results show significant gains (17% gain in average performance) for CLUTRR across different test sets along with superior systematic generalization and competitive results on SWQ, (c) Interpretability, by showing qualitative results of the learnt rules.

## 2 Related Work

Relation linking has been important for various NLP tasks such as semantic parsing, knowledge graph induction (Gardent et al., 2017; Chen et al., 2021; Rossiello et al., 2022; Lin et al., 2020) and knowledge base question answering (Rossiello et al., 2021; Kapanipathi et al., 2020; Neelam et al., 2022). Prior to the surge of generative models, relation linking was addressed either by graph traversal based (Pan et al., 2019; Dubey et al., 2018) or by linguistic-features based methodologies (Sakor et al., 2019a,b; Lin et al., 2020). Several learning based approaches to relation linking have been proposed (Mihindukulasooriya et al., 2020; Yu et al., 2017; Bornea et al., 2021). Most recent approaches to relation linking have focused on generative models (Rossiello et al., 2021). These approaches are data hungry and non-interpretable. In contrast, our work is a self-supervised rule learning based approach for relation linking. The learnt rules are

---

[3]Triple format: *predicate(source entity, destination entity)*, where *source entity* and *destination entity* are also alternatively referred to as *subject* and *object*, respectively.

human readable and they can be learnt for different knowledge bases as context for different domains.

Rule learning, specifically in terms of detecting relations has also been popular for knowledge base completion (KBC) task. KBC problems are addressed in two different ways in the literature. Rule based KBC learns explainable rules that can be used to predict the missing links in the knowledge base (Sen et al., 2021; Qu et al., 2020; Yang et al., 2017a; Sadeghian et al., 2019; Purgal et al., 2022; Rocktäschel and Riedel, 2017). These rules are learned for predicates in the knowledge base in terms of other predicates and paths in the knowledge base. Our work uses the rule model and some of the ideas from the rule based KBC to build self supervised explainable rules for linking text elements to knowledge base relations. There have also been several learning based methods that use vector embeddings of the entities and relationships and then use them to predict any missing relationships (Nickel et al., 2015; Yang et al., 2014; Wang et al., 2014; Lin et al., 2015; Trouillon et al., 2016; Sun et al., 2018; Zhang et al., 2019); they have the same drawbacks as neural relation linking approaches. There are many recent works that use semi-supervised/unsupervised learning for rules and knowledge induction (Pryzant et al., 2022; Bhagavatula et al., 2023; Lang and Poon, 2021; Zhu and Li).

Transformers (Vaswani et al., 2017) have given rise to a wide range of masked pre-training models that try to learn latent representation of words. Masked language modeling has gained popularity in building large pre-trained models that learn word representations that can easily be adapted to a variety of tasks. Masked language training has resulted in building various model architectures like encoder only models which learn a vector representation of tokens using masked training (Peters et al., 2018; Devlin et al., 2019; Liu et al., 2019; Clark et al., 2020). Several Encode-decoder LLMs are built using similar training strategy (Song et al., 2019; Lewis et al., 2019; Soltan et al., 2022; Raffel et al., 2019). Recently decoder only models built using similar masked language training are gaining popularity (Radford and Narasimhan, 2018; Radford et al., 2019; Brown et al., 2020; Chowdhery et al., 2022; Zhang et al., 2022; Scao et al., 2022). In this paper we take inspiration from masked language training on text and use the ideas to perform similar masking on logic statements to learn rules.

# 3 Our Approach

Our approach to relation linking is a rule-learning framework where rules learned in a self-supervised fashion are used to map from 'NL text segments' to the 'elements of structured knowledge'. For self-supervised rule learning, we use masked training. A distinctive aspect of the masked training in our approach is that it is applied over the logical form of the input text. When logical form is not readily available, we first convert NL text into its logical form using a semantic parser, before applying masked training. In this section, first we briefly describe the conversion of input text to logical statements, and then describe the proposed rule model and its self-supervised training.

## 3.1 Logical Statements

In this work, we use triples extracted from sentences as their logical form[4]. Note that logical forms are expressed as *predicate(source entity, destination entity)*, as illustrated in Figure 1. To extract such logical forms from NL text, we use the semantic parsing approach described in (Neelam et al., 2022), which performs an Abstract Meaning Representation (AMR) (Banarescu et al., 2013) based $\lambda$-expression extraction. For simple sentences, $\lambda$-expressions can be converted directly into triples[5]. Figure 1 shows an illustration of a NL text and its corresponding logical form. Different elements of the triple represent distinct segments of the NL text. Since we use an AMR based approach to derive $\lambda$-expressions, predicate components of the triple are typically a propbank predicate. Relation linking is expected to map this predicate component onto the corresponding knowledge-base elements.

## 3.2 Rule Model

Towards the goal of relation linking, our rule model is formulated as a function that maps a weighted mixture of 'knowledge base predicates' onto a 'predicate component of the NL text'. This enables interpretable prediction of the set of knowledge base predicates that the text predicate should get linked to. Figure 2 gives an illustration of a simple rule.

Let us assume a triple $pred_i(s, d)$ extracted from NL text, where $s$ is the source entity, $d$ is the destination entity and $pred_i$ is the text predicate. The

---

[4]We use triples and logical form interchangeably.
[5]We also employ heuristics to handle the AMR and $\lambda$-expression computation failures and errors.

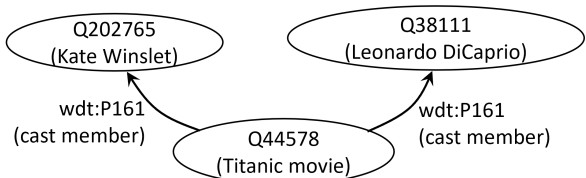

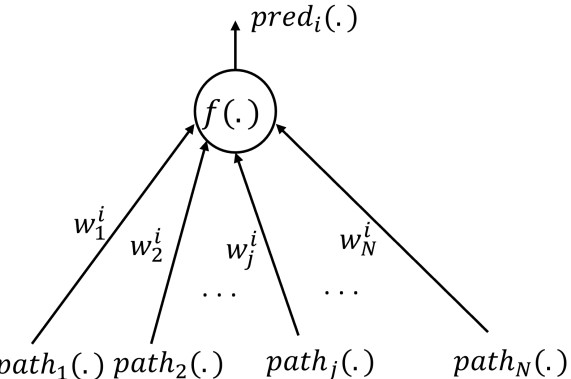

Figure 3: Illustration of a one-hop path between entities Kate Winslet and Leonardo DiCaprio via entity Titanic movie in Wikidata (through a reverse relation edge *cast-member* connecting *Kate Winslet* to *Titanic movie* and a forward relation edge *cast-member* connecting *Titanic movie* to *Leonardo DiCaprio*). This can also be viewed as an extended context extracted from knowledge base through the unmasked elements of the masked logical statement in Figure 1.

Figure 4: Neural model implementing each rule of the rule model defined in Equation (1).

goal is to ground $pred_i$ to its corresponding knowledge base predicates using relation linking model. Our proposed rule model to achieve this is defined as:

$$pred_i(s, d) = f\left(\sum_{j=1}^{N} w_j^i \ path_j(s, d)\right) \quad (1)$$

for $i \in \{1, \ldots, M\}$ where $f(\cdot)$ denotes a function such as Sigmoid and $path_j$ denotes a path in the knowledge base that passes through one or more knowledge base predicate edges. $path_j(s, d)$ can take values 0 or 1 depending upon presence or absence of a path $path_j$ connecting source entity $s$ and destination entity $d$. Depending upon the length of the path, i.e., number of predicate edges and the intermediate entities, paths could be categorized as $k$-hop, where $k$ denotes the count of intermediate entities in the path. An example of a 1-hop path between entities in Wikidata is shown in Figure 3. If there are $L$ unique knowledge base predicates, i.e., $\{p_l\}, \ 1 \leq l \leq L$, the sets of 0, 1 and 2-hop paths are defined as follows:

$$\text{0-hop paths} = \{p_l(s, d)\}, \ 1 \leq l \leq L$$

$$\text{1-hop paths} = \{p_l(s, *) : p_m(*, d),$$
$$1 \leq l \leq L, \ 1 \leq m \leq L$$

$$\text{2-hop paths} = \{p_l(s, *) : p_m(*, *) : p_n(*, d)\},$$
$$1 \leq l \leq L, \ 1 \leq m \leq L, \ 1 \leq n \leq L$$

where $*$ denote any possible entity. Note that paths always originate from a source entity and end at a destination entity. Also note that, the constituent knowledge base predicates in a path could also be inverse (reverse), denoted by superscript $^{-1}$. For example, path in Figure 3 contains an inverse relation,

which is denoted as *cast member*$^{-1}$ in the corresponding path expression given in Figure 2. All the knowledge base paths of various lengths, i.e., from 0-hop paths until $k$-hop paths put together, constitute the overall set of paths considered in our rule learning framework. Let us assume there are $N$ such paths in total. Given that the knowledge base paths $path_j$ are defined in terms of the knowledge base predicates, our rule model in (1) effectively defines a mapping between the 'NL text predicates' and the 'knowledge base predicates'. During learning, we want to estimate weights $w_j^i$ so that rule model in (1) can be used for relation linking.

A neural implementation of each rule in (1) is given in Figure 4. Note that there is one such network each for every rule in (1), and all these networks are trained together during learning. Note that the parameters of our rule model, denoted by $W$, is a matrix of learnable weights:

$$W = \{w_j^i\}, \ 1 \leq i \leq M, \ 1 \leq j \leq N$$

where each row corresponds to the set of path weights of a specific rule. In the next section, we describe masked training used to estimate these weights in a self-supervised manner.

### 3.3 Self-Supervised Rule Learning

Masked training in large language model training involves masking certain parts of the input text and having the model predict those masked parts by treating the rest of the (unmasked) text as its context. The model parameters are adjusted during training to effectively model the context so it can act as a proxy for the masked part. In our approach masked training is performed on the logical form of the input text as detailed below.

### 3.3.1 Masked Training on Logical Forms

Logical statements are composed of distinct elements playing distinct semantic roles. For example, triples are composed of a predicate, a source entity (also called subject) and a destination entity (also called object). The masking procedure could mask any of these elements:

1. Predicate masking - predicate component of the triple is masked

2. Source entity masking - source entity component of the triple is masked

3. Destination entity masking - destination entity element of the triple is masked

Figure 1 gives an illustration of predicate masking. Similar to the large language model training, masked training for rule learning also aims to adjust the parameters of the rule model so that it can accurately predict the masked elements from the unmasked context.

However, the distinctive aspect of our approach is in the context computation. For the context, in addition to the unmasked elements of the logical statement, we also use paths within the knowledge base that are associated to those unmasked elements. A set of such associated paths are obtained by grounding the unmasked elements of the logical statement to the knowledge base. For example, given Wikidata as the knowledge base, we first use entity linking (Wu et al., 2019) to link the unmasked source and destination entities of the triple to corresponding Wikidata entities, as illustrated in Figure 1. Then the Wikidata paths originating from unmasked source entity and ending at unmasked destination entity are computed and used as the context. Figure 3 gives an illustration of a Wikidata path, through entity node corresponding to *Titanic movie*, for the unmasked entities of Figure 1. Note that such use of knowledge base paths as the context help establish the mapping between the NL text predicates and the knowledge base predicates, which indeed is the goal of our relation linking model. Next we describe rule model parameter estimation procedure for different masking scenarios.

### 3.3.2 Rule Model Estimation

Learning procedures for the 3 masking scenarios differ mainly in the way training loss is computed. In all 3 masking scenarios we use rule models to predict the masked element. As discussed

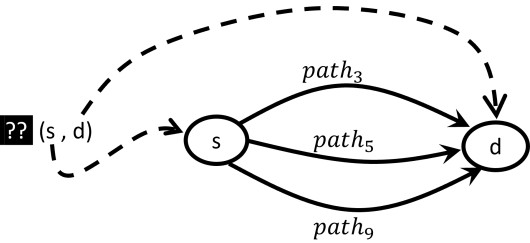

Figure 5: Illustration of knowledge base paths to be used as context when predicate in the triple is masked.

above, knowledge base paths associated with the unmasked elements of the triple are used as the context for the prediction. However, the set of such paths differ based on which element is masked and which elements are unmasked and could be linked to the knowledge base. Training loss functions are defined to improve prediction accuracy over the training iterations, as described below:

**a) Predicate Masking**: As illustrated in Figure 5, when the predicate is masked, unmasked source and destination entities are linked to the corresponding elements in the knowledge base through entity linking. Then the set of all paths that connect these linked entities are computed and used as context. Let $j \in E$ denote the set of indices of knowledge base paths that connect the linked source and destination entities. These paths when applied to all the rules in (1), corresponding to all the text predicates $pred_i(.), \ 1 \leq i \leq M$, would result in scores for each text predicate as below:

$$s_i = f\left(\sum_{j \in E} w_j^i \ path_j(s, d)\right), \ 1 \leq i \leq M.$$

For accurate prediction of the masked predicate by the model, the score should be highest (1.0) for the text predicate being masked and should be the lowest (0.0) for all others. Accordingly, let $t_i, \ 1 \leq i \leq M$ denote the target scores, where the score corresponding to the index of the masked predicate is 1.0 and the rest of the scores are 0.0. We compare the estimated scores against these target scores, to compute the training loss that could be used further to update the model weights through stochastic gradient descent (SGD). In actual, we use a cross-entropy training loss computed as below:

$$loss = \sum_{j=1}^{N} t_i \ log(s_i)$$

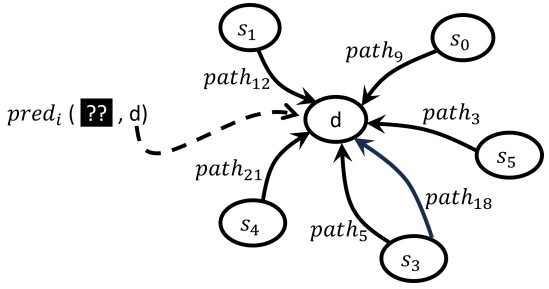

Figure 6: Illustration of knowledge base paths to be used as context when source entity in the triple is masked.

**b) Source Entity Masking**: As illustrated in Figure 6 when the source entity is masked, unmasked destination entity is linked to the corresponding element in the knowledge base, through entity linking. Then the set of all paths that end at the linked destination entity are computed for use as the context. Note that all these paths together have a set of associated origination entities that could potentially be considered as the prediction for the masked source entity. Let $S = \{s_1, s_2, ..., s_E\}$ denote the set of all such candidate source entities. Let $E_e$ correspond to the set of indices of all paths that originate from the candidate source entity $s_e$ and end at the unmasked linked destination entity. These paths when applied to a specific rule in (1) that corresponds to the unmasked text predicate $pred_i$, would give a score for $s_e$ as below:

$$s_e = \sum_{j \in E_e} w_j^i \ path_j(s_e, d), \ \ 1 \le e \le E$$

A list of such scores are computed for each candidate source entity in $S$. Note that among these scores the one corresponding to candidate entity that is same as the masked source entity should be highest (1.0) and the score for the rest should be the lowest (0.0). Accordingly, target scores are $t_e, \ 1 \le e \le E$, where the score for the index of the masked entity is set to $1.0$ and the rest are set to $0.0$. We then compute training loss by comparing the estimated scores against the target scores. A cross-entropy training loss computed as below is used to update the rule model weights through stochastic gradient descent (SGD):

$$loss = \sum_{e \in S} t_e \ log(s_e)$$

**c) Destination Entity Masking**: Destination entity masking is similar to that of the source entity masking, except that when the destination entity is masked, unmasked source entity is linked to the knowledge base and the set of all paths originating from the linked source entity are computed as context, resulting in the associated candidate destination entities. Then scores for all candidate destination entities are computed in a similar fashion to further compute the training loss.

### 3.4 Inference

Note that, once trained, the right hand side of the rule model in Equation (1) directly gives human readable information of the relation linking, highlighting the interpretability/explainability aspect of our model. Each rule gives information of the knowledge base paths (that in turn are composed of the knowledge base predicates) that the text predicate should get linked to, with path weights denoting the linking confidence. During inference, for example while using our rule model to perform QA, the information being asked in the NL question would correspond to the missing element in the corresponding logical form. Hence the goal is to estimate candidates for that missing element and score them using our rule model. The scoring procedure is similar to that for the masked elements during masked training as described in Section 3.3.2. Top-K scoring candidates are chosen as the answers estimates of the rule model.

### 3.5 Scaling

The number of paths $N$ is exponential in the number of predicates $L$ that exist in the knowledge base. As a result, solving problem (1) becomes prohibitive with large $N$ and hence we explore methods for reducing the search space of paths. In other words, we learn a set of paths $\mathcal{P}$ with $|\mathcal{P}| << N$ that contains the optimal paths, allowing us to replace the summation in problem in Equation (1) to a summation over $\mathcal{P}$, and thereby reducing the computation required to find the solution.

We make use of a Chain of Mixtures (CoM) model, also known as Neural Logic Programming (Yang et al., 2017b) or edge-based rules (Sen et al., 2022), to learn $\mathcal{P}$. In CoM, a rule is represented by a conjunction of mixtures of predicates. Specifically, rather than define a weight per path, for each of $M$ predicates we define a weight per predicate per hop to get rules of the form:

$$pred_i^{CoM}(s, d) = g\left(f\left(\sum_{j=1}^{M} w_{j,0}^i p_j(s, r_1)\right),\right.$$
$$\left. f\left(\sum_{j=1}^{M} w_{j,1}^i p_j(r_1, r_2)\right),\right. \tag{2}$$

$$f\left(\sum_{j=1}^{M} w_{j,2}^{i} p_j(r_2, d)\right)$$

for $i \in \{1, \ldots, M\}$ where now $w_{j,k}^{i}$ is a weight for rule $i$ (i.e., for the $i^{th}$ predicate) for predicate $j$ on the hop $k$ of the rule being learned, and $g(\ldots)$ is a function approximating a conjunction such as a product. Equation (2) is an example 2-hop rule where $r_1$ and $r_2$ can take the values of any entities in the knowledge base. The number of weights learned in a CoM rule is $M(K+1)$ where $K$ is the number of hops in the rule, which greatly reduces the number of weights learned in problem (1).

While CoM requires learning much fewer parameters than our rule model (1), it is less interpretable because it is not known which paths are actually relevant to the rule. CoM rules suggest that any path that can be obtained from predicates with positive weights in the hops is possible. For example, suppose there are 8 predicates in the knowledge base and the set of strictly positive weights in a rule of the form (2) is $\{w_{0,1}^{i}, w_{0,3}^{i}, w_{0,4}^{i}, w_{1,1}^{i}, w_{1,2}^{i}, w_{2,2}^{i}, w_{2,4}^{i}, w_{2,8}^{i}\}$. Then such a chain rule only allows rules of the form $p_j(s, r_1) \wedge p_k(r_1, r_2) \wedge p_l(r_2, e)$ where $j \in \{1, 3, 4\}$, $k \in \{1, 2\}$, $l \in \{2, 4, 8\}$. with 8 predicates, there are a total of $8 \times 8 \times 8 = 512$ possible paths, but the CoM reduces this to $3 \times 2 \times 3 = 18$ possible paths. Given these possible paths, a more interpretable rule of the form Equation (1) can be learned more efficiently since the summation can be taken over 18 paths instead of $N = 512$ paths.

## 4 Experimental Setup

### 4.1 Datasets

We evaluate our approach on two datasets as described below that involve utilization of two distinct types of knowledge sources.

**a) CLUTRR** (Sinha et al., 2019) is a benchmark dataset for evaluating the reasoning capabilities of a Natural Language Understanding (NLU) system, where the goal is to infer kinship relations between source and target entities based on short stories. It contains both graph and textual representations of these stories. Each story consists of clauses (chains of relations) of length ranging from 2 to 10. The benchmark tests the model's ability for: *i) Systematic Generalization* which tests how well the model generalizes beyond the combination of rules seen during training. The test set consists of clauses

of length up to 10, while the train set has clauses of length up to 4. *ii) Model robustness* which tests how well model performs in the presence of noisy facts. It consists of three kinds of noise: a) *Supporting facts:* It adds alternative equivalent paths between source and target entities. b) *Irrelevant facts:* It adds a path which originates on the path between the source and target entity but ends at an entity not in that path. c) *Disconnected facts:* It adds a path which is completely disconnected from the path between the source and target entity.

We considered graph stories with only clause length 2 for training rule models in a self-supervised fashion. This results in multiple one-hop rules along with their confidence scores. These rules can be used to handle stories of clause length 2 at the time of inference. For stories with clause length greater than 2, we combine relations within the path from source to target entity using Algorithm 1 to compute the path scores. A n-length path $R_1, ..., R_n$ from source entity to target entity can be reduced by invoking *infer(1, n)*. It utilises two sub-modules *predict($R_i$, $R_j$)* and *score($R_i$, $R_j$)* which returns prediction and confidence scores for merging $R_i$ and $R_j$ using one-hop rule model.

---

**Algorithm 1:** Inference procedure using one-hop rule model

**Output :** $R_{ij}$: Prediction for path $R_i$ ...$R_j$,
$\qquad$ $S_{ij}$: Prediction score for path $R_i$ ...$R_j$
**Function** infer($i, j$):
$\quad$ **if** $i == j$ **then**
$\quad\quad$ | return $R_i$, 1
$\quad$ **else**
$\quad\quad$ $R_{ij} \leftarrow$ null, $S_{ij} \leftarrow 0$
$\quad\quad$ **for** $k \leftarrow i + 1$ **to** $j$ **by** 1 **do**
$\quad\quad\quad$ $R_{ik-1}, S_{ik-1} \leftarrow infer(i, k-1)$
$\quad\quad\quad$ $R_{kj}, S_{kj} \leftarrow infer(k, j)$
$\quad\quad\quad$ $S'_{ij} \leftarrow S_{ik-1} * S_{kj} * score(R_{ik-1}, R_{kj})$
$\quad\quad\quad$ $R'_{ij} \leftarrow predict(R_{ik-1}, R_{kj})$
$\quad\quad\quad$ **if** $S'_{ij} > S_{ij}$ **then**
$\quad\quad\quad\quad$ | $R_{ij} \leftarrow R'_{ij}, S_{ij} \leftarrow S'_{ij}$
$\quad\quad$ **end**
$\quad\quad$ **return** $R_{ij}, S_{ij}$
$\quad$ **end**

---

**b) SWQ-WD** (Diefenbach et al., 2017) is a Question Answering (QA) dataset with simple questions built to evaluate the QA accuracy of KBQA systems that use Wikidata as the knowledge source. This dataset fits our relation linking evaluation goal because it consists of simple questions with single text predicates, hence it is possible to perform a focused evaluation of the relation linking task, reducing the impact of uncertainties of other components of KBQA systems such as semantic parsing

**CLUTRR:**

- **son**: { (son **and** brother, 0.95), (daughter **and** brother, 0.95), (wife **and** son, 0.96), (husband **and** son, 0.96) }
- **granddaughter**: { (son **and** daughter, 0.97), (daughter **and** daughter, 0.96),
  (granddaughter **and** sister, 0.92), (grandson **and** sister, 0.96),
  (wife **and** granddaughter, 0.94), (husband **and** granddaughter, 0.90) }
- **niece**: { (brother **and** daughter, 0.98), (sister **and** daughter, 0.91) }

**SWQ-WD:**

- **book-write**: { (author, 0.88), (notable work$^{-1}$, 0.87) }
- **film-language**: { ... (original language of film or TV show, 0.97), (language of work or name, 0.83) , ...,
  (cast member **and** writing language, 0.97), (screenwriter **and** writing language, 0.95),
  (film editor **and** languages spoken, written or signed, 0.87),
  (director **and** languages spoken, written or signed, 0.83),
  (follows **and** original language of film or TV show, 0.96),
  (based on **and** language of work or name, 0.70), .... }

Figure 7: Sample learned rules for both CLUTRR and SWQ-WD datasets. Note that each rule is represented by a set of tuples, with each tuple containing a knowledge base path and its weight value. These corresponds to the knowledge base paths and the corresponding weights in rule model (1). Note that inverse knowledge base predicate (i.e., reverse knowledge base relations) in the rules are denoted by superscript $^{-1}$, as described in Section 3.2.

- **son:** { (son, 0.97), (husband , 2.49), (daughter , 0.70), (wife , 2.30)} **and** {(son , 0.81), (brother , 2.86) }
- **granddaughter:** { (son, 1.26), (grandson, 2.50), (daughter, 1.20), (granddaughter, 4.22)}
  **and** {(daughter, 1.22), (granddaughter, 5.09) }
- **niece:** { (brother, 1.69), (sister, 1.43)} **and** {(daughter, 1.72) }

Figure 8: Sample learned rules for CLUTRR using Chain of Mixtures.

and entity linking. SWQ-WD has 13894 train set questions and 5622 test set questions. We use train set to self-learn rules and report relation linking accuracy on the test set. Answer prediction using our rule model is described in Section 3.4.

## 4.2 Baselines and Metrics

For both the datasets, we compared our method against the best available baseline as per our knowledge. For CLUTRR, we compared against GAT (Veličković et al., 2017), a neural graph representation framework that learns representations through self-attention over the neighbourhood. For SWQ, we compared against SYGMA (Neelam et al., 2022), a modular and generalizable pipeline based approach consisting of KB-specific and KB-agnostic modules for KBQA.

For CLUTRR experiments, where relation linking performance is directly evaluated, we use a simple accuracy metric of $accuracy = success\_count/total\_count$. For SWQ experiments, where the relation linking performance is not directly evaluated but through its influence on the question answering (QA) accuracy, we use the metric of $Recall$ which is a fraction of correct answers that the system is able to estimate.

## 4.3 Code Release

The code and the experimental setup used to train and evaluate our self-supervised rule learning based approach for relation linking is available at https://github.com/IBM/self-supervised-rule-learning.

## 5 Results

**a) CLUTRR:** Figure 7 shows (in upper part) sample 1-hop rules learned from CLUTRR dataset. Note that these rules are of form as in Equation (1). For CLUTRR, there were 22 predicates and so each 1-hop rule was learned over $N = 22 \times 22 = 484$ possible paths. Figure 8 shows corresponding CoM rules for the CLUTRR dataset. For example, the CoM rule for *son* implies possible 1-hop rules that start with either *son*, *husband*, *daughter*, or *wife* and end with either *son* or *brother*, offering $4 \times 2 = 8$ possible rules for *son*. Rules of the form (1) could thus be rather learned over only these 8 possible rules and capture the same two interpretable rules for *son* seen in the CLUTRR part of Figure 7. Overall, rules for the 22 different relations search over 484 paths each (10648 total possible rules); CoM reduced that search to 118 to-

tal possible rules with perfect coverage (i.e., CoM results include all optimal rules as a subset) for all but 2 relations.

Figure 9 compares the *Systematic generalization* performance of our approach against GAT (Veličković et al., 2017) on CLUTRR dataset with test set clause length ranging from 2 to 10. The results suggest that the proposed approach trained with stories of clause length k = 2 generalizes better in comparison to GAT trained with stories of clause length k = 2, 3 and stories of clause length k = 2, 3, 4. Table 1 compares the *model robustness* of our approach against GAT on CLUTRR dataset for differing noise scenarios during training and inference. The results suggest that the proposed approach outperforms GAT for most scenarios, achieving an average absolute accuracy improvement of 17%.

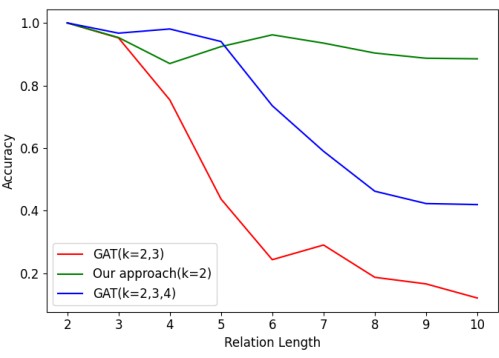

Figure 9: Systematic generalization on CLUTRR.

**b) SWQ-WD:** Figure 7 shows (in lower part) sample 0/1-hop rules learned from SWQ dataset, of form as in Equation (1). Interestingly, for text predicate *film-language* it is able to learn rules that associate it with the language spoken/written by the director or screen writer or cast members of the film. Table 2 compares QA accuracy of our approach against SYGMA (Neelam et al., 2022). Our approach achieves comparable QA performance (as given in the first two lines), in spite of SYGMA using supervised training for its relation linking module - whereas our approach is a self-supervised approach (not using any labelled data for training). Note that rule learning is prone to inaccurate entity linking[6]. Thus, when ground truth entities are

---

[6]Note that inaccuracies both in AMR parsing and entity linking could lead to erroneous logical statements. However, SWQ sentences are simple and hence their AMR parsing is straight forward, with SMATCH score of 83.0% (Neelam et al., 2022). Thus, incorrect logical statements for SWQ are largely due to entity linking errors. In case of CLUTRR, GAT baseline used the readily available ground truth logical forms of the sentences that we also used for fair comparison.

| Training | Testing | GAT | Our approach |
|---|---|---|---|
| Clean | Clean | **1.0** | **1.0** |
| | Supporting | 0.24 | **1.0** |
| | Irrelevant | 0.51 | **1.0** |
| | Disconnected | 0.8 | **1.0** |
| Supporting | Clean | **0.92** | 0.89 |
| | Supporting | **0.98** | 0.97 |
| | Irrelevant | 0.5 | **0.94** |
| | Disconnected | **0.92** | 0.91 |
| Irrelevant | Clean | 0.92 | **0.94** |
| | Supporting | 0.77 | **0.94** |
| | Irrelevant | 0.93 | **0.96** |
| | Disconnected | 0.85 | **0.88** |
| Disconnected | Clean | 0.75 | **0.91** |
| | Supporting | 0.78 | **0.93** |
| | Irrelevant | 0.56 | **0.91** |
| | Disconnected | **0.96** | 0.93 |
| Average | | 0.77 | **0.94** |

Table 1: Model robustness on CLUTRR - Accuracies for various training and inference noise conditions.

| Method | Recall |
|---|---|
| SYGMA | **55.0** |
| Our approach | 53.7 |
| SYGMA (with GT entities) | 68.0 |
| Our approach (with GT entities) | **70.1** |

Table 2: QA performance on SWQ.

used, our approach performs marginally better than SYGMA (as shown in the last two rows). Note that when ground truth entities are used, QA performance is more reflective of the relation linking performance, thus asserting the effectiveness of our relation linking model.

## 6 Conclusions

In this paper, we presented a novel masked training based self rule learning approach for relation linking. The proposed approach takes inspiration from the masked training of large language models and rule learning in knowledge base completion to create a rule learning model that results in a relation linking model that is interpretable and generalizable to any knowledge base and domain. Our evaluation on two datasets CLUTRR (rule learning for kinship relations) and SWQ (relation linking for Wikidata) shows that our model is able to produce high quality rules for relation linking that generalize across domains. Our next steps are to scale to learn from large amount of data, extend the framework beyond triples. We see a lot of potential for our rule learning framework beyond relation linking, like general knowledge acquisition.

## 7 Limitations

Proposed approach is an initial attempt at taking masked training to logic statements and using knowledge graph as context. Current approach described is only taking single triple to learn relationship linking problem for KBQA, but overall approach has potential to extend to larger logic statements. Extending the work to handle logic statements with more than one triple is a future direction that we are looking into. In the current approach we assumed semantic parses to be perfect and available, but in reality they can be noisy resulting in noisy rule model. Handling noise coming from the parsing is another main extension we would like to look into as part of our future work.

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
