# OpenReview forum: "Self-Supervised Rule Learning to Link Text Segments to Relational Elements of Structured Knowledge"
_EMNLP/2023/Conference — EMNLP 2023 Findings_

### Official Review · Reviewer_aFLf · 2023-08-03

**Soundness:** 3

**Excitement:**

3: Ambivalent: It has merits (e.g., it reports state-of-the-art results, the idea is nice), but there are key weaknesses (e.g., it describes incremental work), and it can significantly benefit from another round of revision. However, I won't object to accepting it if my co-reviewers champion it.

**Paper Topic And Main Contributions:**

In this paper, the authors present an approach to self-learn rules that serve as interpretable knowledge to perform relation linking in knowledge base QA systems.

**Questions For The Authors:**

- the advantages of transforming link text segments into relational elements of structured knowledge are not sufficiently detailed:
1) using the authors' solution how much can the quality of a QA system be increased?
2) Have the authors made experiments where to use their solution in a wider context? (such as a QA-type system) If so, how much has the quality of the previous solution increased? What are the advantages of using the solution proposed by them in this paper in such a system?

**Reasons To Accept:**

- good and actual references from important conferences and journals
- the formalization of the problem is well done
- the examples help the reader to understand better what is the problem and how the authors deal with it
- the quality of the obtained results which outperforms GAT demonstrates the robustness of the proposed method

**Reasons To Reject:**

- in subsections from section 3 the authors need to add more examples to help the reader to understand how the proposal method works
- there are also some questions without answers in the paper (see below section)

**Reproducibility:**

3: Could reproduce the results with some difficulty. The settings of parameters are underspecified or subjectively determined; the training/evaluation data are not widely available.

**Reviewer Confidence:**

3: Pretty sure, but there's a chance I missed something. Although I have a good feel for this area in general, I did not carefully check the paper's details, e.g., the math, experimental design, or novelty.

**Typos Grammar Style And Presentation Improvements:**

Line 452: "l ∈ {2,4,8}." => "l ∈ {2,4,8},"
Line 511: "13894" => "13,894"
Line 512: "5622" => "5,622"
Line 531: "10648" => "10,648"

---

> ### Author Rebuttal · Authors · 2023-08-29
>
> We thank the reviewer for detailed comments. Please find answers to the questions raised in the review below
>
> **Response to 'Reason To Reject':**
>
> Due to space constraints, we could not add more examples. In the revision, we can add more examples to improve the overall readability of the paper using the additional page of content allowed in the final version. We will address the typos in the final version. Regarding reproducibility, we will open source the code and add a git link to the final version.
>
>
> **Response to 'Questions For The Authors':**
>
> **the advantages of transforming link text segments into relational elements of structured knowledge are not sufficiently detailed:**
>
> The role of relation linking in QA is well studied and there are many works [e.g., SYGMA] which show its importance in QA system.  We have also described a body of work in the first paragraph of the related work section that motivates relation linking. One of the critical components of any KBQA system is relation linking along with the entity linking. Entity linking accuracy is generally in the high 90s leaving relation linking a critical component that has the maximum effect on the overall QA. The SYGMA paper along with other relation linking papers provide ample evidence as to how critical the relation linking task is. We can add numbers from the references that correlate relation linking performance and overall QA performance to show the value of relation linking.
>
> **1 & 2. using the authors' solution how much can the quality of a QA system be increased? & Have the authors made experiments where to use their solution in a wider context? (such as a QA-type system) If so, how much has the quality of the previous solution increased? What are the advantages of using the solution proposed by them in this paper in such a system?**
>
>  Table 2 gives the recall performance for QA on SWQ compared using our rule learning approach for relation linking. The key advantages of our relation linking with self supervised rule learning is that we do not require any supervised data and that we get human readable interpretable rules that can help in explaining the predictions in practical applications.

---

### Official Review · Reviewer_CrYn · 2023-08-05

**Soundness:** 2

**Excitement:**

4: Strong: This paper deepens the understanding of some phenomenon or lowers the barriers to an existing research direction.

**Paper Topic And Main Contributions:**

The paper proposes a novel approach called "Self-Supervised Rule Learning to Link Text Segments to Relational Elements of Structured Knowledge" to address the shortcomings of neural approaches in relation linking for knowledge base question answering systems. The main problems tackled are data hunger, lack of interpretability, and limited generalization to other domains.

The proposed approach models relation linking as a self-supervised rule learning task. It leverages the masked training strategy commonly used in large language models but operates on logical statements derived from the sentences rather than raw text inputs. By doing so, the approach extracts extended context information from the structured knowledge source, enabling the construction of more robust and interpretable rules. These rules define natural language text predicates as weighted mixtures of knowledge base paths, and the learned weights effectively map text segments to the corresponding relational elements in the knowledge base.

Main Contributions:
1. Rule Learning for Relation Linking: The paper introduces relation linking as a rule learning task in a self-supervised setting. This novel approach enables better generalization across different domains and reduces the need for extensive annotated data compared to traditional neural methods.
2. Interpretable Relation Linking: Unlike neural network-based approaches, the proposed method offers interpretability based on the learned rules. This provides valuable insights into how the relation linking decisions are made.
3. Comprehensive Evaluation: The paper includes a thorough evaluation of the proposed approach on two datasets, CLUTRR (for rule learning related to kinship relations) and SWQ (for relation linking in Wikidata). The results demonstrate the effectiveness of the approach in producing high-quality rules that generalize well across domains.


**Questions For The Authors:**

A: How accurate is the system (semantic parser) that generates logical forms from the sentences? How does its performance affect the presented methodology?
B: Section 3.3, is `path_j(sad)` a binary variable (line 224) or the length of the path between the entities?
C: In Section 3.2.2 b), if `Ee` is the set of all paths that end in d, it appears that the masked source is the only remaining source candidate, potentially making the task trivial. Can you clarify this aspect and explain how the approach addresses such cases?
D: Can the model predict cases where "no relationship" exists? If not, how are these cases handled in the relation linking process?
E: Could you elaborate on how the context is used during training time?


**Reasons To Accept:**

- The paper presents a novel approach to relation linking, modeling it as a self-supervised rule learning task. This is a fresh perspective on the problem and can offer new insights and benefits compared to traditional neural methods.
- The paper explicitly identifies the shortcomings of neural approaches in relation linking, such as data hunger, lack of interpretability, and limited generalization to other domains. By proposing a self-supervised rule learning approach, the paper aims to overcome these limitations.
- The paper claims that the self-supervised rule learning approach enables better generalization and requires fewer annotated data compared to other methods.
- The approach offers interpretable results by generating rules that define natural language text predicates as weighted mixtures of knowledge base paths. This can provide valuable insights into how the relation linking decisions are made and contribute to the transparency of the model.


**Reasons To Reject:**

- The methodology shows a high level of intricacy and needs a more lucid exposition. Multiple readings may be required to grasp the technical details, which can hinder the paper's accessibility and understanding for the broader community.
- While the paper claims to offer interpretable results, it lacks an example describing what the learned rules represent. Providing concrete examples would be crucial to demonstrate the interpretability of the proposed approach.
- The paper does not provide a clear explanation of how the logical forms from the sentences are generated, and the accuracy of this process remains unknown. Understanding the process of generating logical forms is vital to assess the reliability of the proposed approach.
- The paper needs to provide a clearer explanation of the relationship between paths, rules, and relation linking. A more coherent description would enhance the readers' understanding of the proposed approach

In general, the presentation is insufficient to properly convey the merits of the paper.


**Reproducibility:**

1: Could not reproduce the results here no matter how hard they tried.

**Reviewer Confidence:**

2: Willing to defend my evaluation, but it is fairly likely that I missed some details, didn't understand some central points, or can't be sure about the novelty of the work.

**Typos Grammar Style And Presentation Improvements:**

# Style

- Paragraph starting in line 206 needs better explanation.
- Equation 1, What is `N`?
- Include a formal definition of relation linking

---

> ### Author Rebuttal · Authors · 2023-08-29
>
> **Response to 'Reasons To Reject':**
>
> We thank the reviewer for the detailed comments. We will include more examples to improve overall readability of the paper. Also details on the semantic parser and how logical forms are derived will be added in the final version. We will add clear definitions for paths/rules/relation linking task definitions in the paper. Regarding reproducibility, we will open source the code and add a git link to the final version.
>
> **path**:  path_i(s,d) is nothing but a set of edges between s and d of which by traversing one can reach d from s.  There can be multiple paths that connect two entities in the graph. When we refer to a path_i, we are referring to one of those paths.
>
> **rule** : A rule is nothing but an argmax over weighted paths. Depending on the entities participating in the rules, different paths may be valid and the one with highest score among them is used to predict the masked entity/relation.
>
> **relation linking** : relation linking in general is used in knowledge base context where for a given textual form of the relation in the natural language question, you are expected to predict the corresponding knowledge base relation.  Ex: Leonardo Dicaprio acted in Titanic  then the relation linking goal is to map the text predicate acted in to the P161 (cast member) relationship in wikidata.
>
> **Response to 'Questions For The Authors':**
>
> **A: How accurate is the system (semantic parser) that generates logical forms from the sentences? How does its performance affect the presented methodology?**
>
> AMR is used for parsing the SWQ dataset which has simple sentences with a subject, object, predicate kind of structure. For these sentences, getting the logical form using AMR was straightforward and the SMATCH score for a small test set was approximately 83.0. For the CLUTRR dataset, the ground truth logical forms are readily available with the data and we used those for rule learning similar to the GAT baseline which also uses the same logical forms.
>
> **B: Section 3.3, is path_j(sad) a binary variable (line 224) or the length of the path between the entities?**
>
>  Yes, it is. This means that the model in equation (1) learns a weighted combination of indicators of whether a path exists between s and d using the jth path of predicates.
>
>  **C: In Section 3.2.2 b), if Ee is the set of all paths that end in d, it appears that the masked source is the only remaining source candidate, potentially making the task trivial. Can you clarify this aspect and explain how the approach addresses such cases?**
>
> E_e is the set of all paths that start at some entity s_e and end at d (not the set of all paths that end at d). Suppose there are 10 entities in our knowledge base, and then there would be 10 different sets E_e. For each of these sets (and corresponding possible source entities), a score is computed as in line 371 to use for the training loss in line 381. Note that there can be multiple paths in each E_e, denoted by the two paths from s_3 to d in Figure 6. Suppose castmember(?, Leonardo Dicaprio) is the query. Then castmember can map to a rule that maps multiple paths between destination entity Leonardo Dicaprio through the castmember rule learned during training, resulting in scoring many source entities corresponding to each path and selecting the one with highest score. Please note that the predicate here is not the direct knowledge base predicate which is a single relation, but rather the text predicate which has weighted paths in our rule model.
>
>  **D: Can the model predict cases where "no relationship" exists? If not, how are these cases handled in the relation linking process?**
>
> This would be answered by the predicate masking task used for training in 3.3.2 (a). Specifically, all potential relationships, i.e., predicates, can be scored as in line 341, and no relationship would be indicated by all low scores (or under some defined threshold). As the model is data driven, such a threshold would have to be tuned.
>
> **E: Could you elaborate on how the context is used during training time?**
>
> We refer to context in the paper as the unmasked parts of text samples. Thus, the context can be either a pair of source and destination entities (i.e., in predicate masking), a destination entity and predicate (i.e., in source entity masking), or a source entity and predicate (i.e., in destination entity masking). Then either of the three masking procedures can be done using either of the three corresponding loss functions detailed in 3.2.2. For example, if the context is a pair of source and destination entities, then the loss function for this particular sample will be of the form in line 352 where s_i in line 341 makes use of weights corresponding to the ith predicate. Considering CLUTRR, suppose predicate i in a sample is son(s,d) and let the context in a sample have s=’Adam’ and d =’Jim’.  Then a subset of the paths from those in the top row of Figure 7 might lead to verifying son(Adam, Jim) and this subset corresponds to set E  in the score computed in line 341. Then for this sample the corresponding weights would get updated through the loss in line 352.
>
> **Equation 1, What is N?**
>
> N is the number of overall paths of various lengths (including 0-hop paths through k-hop paths) that can be extracted from a given knowledge base. This is described in the text written after equation (1) in lines 246-250. We will add a comment right after equation (1) to be clear.

---

### Official Review · Reviewer_d5MH · 2023-08-13

**Soundness:** 2

**Excitement:**

4: Strong: This paper deepens the understanding of some phenomenon or lowers the barriers to an existing research direction.

**Paper Topic And Main Contributions:**

The paper proposes a masked training strategy (similar to MLM) over logical forms of sentence to learn rules for relation linking in knowledge bases. The logical forms are of type "predicate(source entity,destination entity)", and one out of predicate/source/destination is masked.
The rule model for relation linking is formulated as a function that maps a weighted mixture of ‘knowledge base predicates’ onto the ‘predicate components of the text'. The rule model is trained in a self supervised manner using the masked training, and the learned rules are human interpretable.
When using this model for inference during QA, depending on the information being asked in the question, the missing element of the logical form of the question is predicted, similar to masked element prediction.
The effectiveness of this method is shown on  CLUTRR dataset (missing kinship relation) and for relation linking in knowledge base question answering system (SWQ-WD).

**Reasons To Accept:**

1) First paper to use mask training on logical forms of text instead of natural text.
2) Proposes a mechanism for self supervised learning of rules for realtion linking, which is useful in the absense of labeled data.
3) The learned rules are human interpretable as shown in the paper.
4) Perfromance on QA is comparable to a spervised learning baseline.

**Reasons To Reject:**

1) The baselines GAT and Sygma are cited but not suffiiciently described in the paper.
2) Abstract Meaning Representation is used to extract logial forms from natural statements. How was the the quality of the extracted forms?
How much does it affect the final perfromance?
3) For both the datsets, only one baseline is presented.
4) Sufficient details for reproducibility are not provided. Will the code be made public?

**Reproducibility:**

3: Could reproduce the results with some difficulty. The settings of parameters are underspecified or subjectively determined; the training/evaluation data are not widely available.

**Reviewer Confidence:**

3: Pretty sure, but there's a chance I missed something. Although I have a good feel for this area in general, I did not carefully check the paper's details, e.g., the math, experimental design, or novelty.

**Typos Grammar Style And Presentation Improvements:**

abstract - "Results demonstrate the effectiveness of our approach" - this line should give more details.

---

> ### Author Rebuttal · Authors · 2023-08-29
>
> We thank the reviewer for the detailed comments. We will improve the presentation issues raised by the reviewer in the final version.
>
> **Response to 'Reasons To Reject':**
>
>
> **1. The baselines GAT and Sygma are cited but not suffiiciently described in the paper.**
>
> We have given references to both the works and will include more details in the revision.
>
> **2. Abstract Meaning Representation is used to extract logical forms from natural statements. How was the the quality of the extracted forms? How much does it affect the final performance?**
>
> AMR is used for parsing the SWQ dataset that has simple sentences with subject, object, predicate kind of structure. For these sentences, getting the logical form using AMR was straight forward and the SMATCH score for a small test set was approximately 83.0. For the CLUTRR dataset, the ground truth logical forms are readily available with the data and we used those for rule learning similar to the GAT baseline which used the same logical forms.
>
> **3. For both the datsets, only one baseline is presented.**
>
> We took the best baseline available as per our knowledge and compared our method with those baselines.
>
> **4. Sufficient details for reproducibility are not provided. Will the code be made public?**
>
> We will open source the code and add a git link to the final version.

---

### Meta-Review · Area_Chair_CRBW · 2023-09-09

**Recommendation:** 2

**Metareview:**

The authors describe a novel approach (similar to masked language model) to self-learn rules that serve as interpretable knowledge to perform relation linking in knowledge base Question Answering.
The paper has some merits and the proposed solution is demonstrated to be effective on 2 datasets.
The main issue of the writing: the proposed method is not clearly described and the paper does not provide enough examples. This lack of clarity does not enable the reader to have a full understanding and obfuscate the quality of the work.

---

### Decision · Program_Chairs · 2023-10-07

**Decision:**

Accept-Findings

**Comment:**

The authors describe a novel approach (similar to masked language model) to self-learn rules that serve as interpretable knowledge to perform relation linking in knowledge base Question Answering.
The paper has some merits and the proposed solution is demonstrated to be effective on 2 datasets.
The main issue of the writing: the proposed method is not clearly described and the paper does not provide enough examples. This lack of clarity does not enable the reader to have a full understanding and obfuscate the quality of the work.